# 3D UX-Net: A Large Kernel Volumetric ConvNet Modernizing Hierarchical Transformer for Medical Image Segmentation

**Ho Hin Lee** [*]
Vanderbilt University

**Shunxing Bao**
Vanderbilt University

**Yuankai Huo**
Vanderbilt University

**Bennett A. Landman**
Vanderbilt University

## Abstract

The recent 3D medical ViTs (e.g., SwinUNETR) achieve the state-of-the-art performances on several 3D volumetric data benchmarks, including 3D medical image segmentation. Hierarchical transformers (e.g., Swin Transformers) reintroduced several ConvNet priors and further enhanced the practical viability of adapting volumetric segmentation in 3D medical datasets. The effectiveness of hybrid approaches is largely credited to the large receptive field for non-local self-attention and the large number of model parameters. We hypothesize that volumetric ConvNets can simulate the large receptive field behavior of these learning approaches with fewer model parameters using depth-wise convolution. In this work, we propose a lightweight volumetric ConvNet, termed 3D UX-Net, which adapts the hierarchical transformer using ConvNet modules for robust volumetric segmentation. Specifically, we revisit volumetric depth-wise convolutions with large kernel (LK) size (e.g. starting from $7 \times 7 \times 7$) to enable the larger global receptive fields, inspired by Swin Transformer. We further substitute the multi-layer perceptron (MLP) in Swin Transformer blocks with pointwise depth convolutions and enhance model performances with fewer normalization and activation layers, thus reducing the number of model parameters. 3D UX-Net competes favorably with current SOTA transformers (e.g. SwinUNETR) using three challenging public datasets on volumetric brain and abdominal imaging: 1) MICCAI Challenge 2021 FLARE, 2) MICCAI Challenge 2021 FeTA, and 3) MICCAI Challenge 2022 AMOS. 3D UX-Net consistently outperforms SwinUNETR with improvement from 0.929 to 0.938 Dice (FLARE2021) and 0.867 to 0.874 Dice (Feta2021). We further evaluate the transfer learning capability of 3D UX-Net with AMOS2022 and demonstrates another improvement of 2.27% Dice (from 0.880 to 0.900). The source code with our proposed model are available at `https://github.com/MASILab/3DUX-Net`.

## 1 Introduction

Significant progress has been made recently with the introduction of vision transformers (ViTs) Dosovitskiy et al. (2020) into 3D medical downstream tasks, especially for volumetric segmentation benchmarks Wang et al. (2021); Hatamizadeh et al. (2022b); Zhou et al. (2021); Xie et al. (2021); Chen et al. (2021). The characteristics of ViTs are the lack of image-specific inductive bias and the scaling behaviour, which are enhanced by large model capacities and dataset sizes. Both characteristics contribute to the significant improvement compared to ConvNets on medical image segmentation Tang et al. (2022); Bao et al. (2021); He et al. (2022); Atito et al. (2021). However, it is challenging to adapt 3D ViT models as generic network backbones due to the high complexity of computing global self-attention with respect to the input size, especially in high resolution images with dense features across scales. Therefore, hierarchical transformers are proposed to bridge these gaps with their intrinsic hybrid structure Zhang et al. (2022); Liu et al. (2021). Introducing the "slid-

---

[*]Correspondence to ho.hin.lee@vanderbilt.edu

ing window" strategy into ViTs termed Swin Transformer behave similarily with ConvNets Liu et al. (2021). SwinUNETR adapts Swin transformer blocks as the generic vision encoder backbone and achieves current state-of-the-art performance on several 3D segmentation benchmarks Hatamizadeh et al. (2022a); Tang et al. (2022). Such performance gain is largely owing to the large receptive field from 3D shift window multi-head self-attention (MSA). However, the computation of shift window MSA is computational unscalable to achieve via traditional 3D volumetric ConvNet architectures. As the advancement of ViTs starts to bring back the concepts of convolution, the key components for such large performance differences are attributed to the **scaling behavior** and **global self-attention with large receptive fields**. As such, we further ask: **Can we leverage convolution modules to enable the capabilities of hierarchical transformers?**

The recent advance in LK-based depthwise convolution design (e.g., Liu et al. Liu et al. (2022)) provides a computationally scalable mechanism for large receptive field in 2D ConvNet. Inspired by such design, this study revisits the 3D volumetric ConvNet design to investigate the feasibility of (1) **achieving the SOTA performance via a pure ConvNet architecture**, (2) **yielding much less network complexity compared with 3D ViTs**, and (3) **providing a new direction of designing 3D ConvNet on volumetric high resolution tasks**. Unlike SwinUNETR, we propose a lightweight volumetric ConvNet 3D UX-Net to adapt the intrinsic properties of Swin Transformer with ConvNet modules and enhance the volumetric segmentation performance with smaller model capacities. Specifically, we introduce volumetric depth-wise convolutions with LK sizes to simulate the operation of large receptive fields for generating self-attention in Swin transformer. Furthermore, instead of linear scaling the self-attention feature across channels, we further introduce the pointwise depth convolution scaling to distribute each channel-wise feature independently into a wider hidden dimension (e.g., $4\times$input channel), thus minimizing the redundancy of learned context across channels and preserving model performances without increasing model capacity. We evaluate 3D UX-Net on supervised volumetric segmentation tasks with three public volumetric datasets: 1) MICCAI Challenge 2021 FeTA (infant brain imaging), 2) MICCAI Challenge 2021 FLARE (abdominal imaging), and 3) MICCAI Challenge 2022 AMOS (abdominal imaging). Surprisingly, 3D UX-Net, a network constructed purely from ConvNet modules, demonstrates a consistent improvement across all datasets comparing with current transformer SOTA. We summarize our contributions as below:

- We propose the 3D UX-Net to adapt transformer behavior purely with ConvNet modules in a volumetric setting. To our best knowledge, this is the first large kernel block design of leveraging 3D depthwise convolutions to compete favorably with transformer SOTAs in volumetric segmentation tasks.

- We leverage depth-wise convolution with LK size as the generic feature extraction backbone, and introduce pointwise depth convolution to scale the extracted representations effectively with less parameters.

- We use three challenging public datasets to evaluate 3D UX-Net in 1) direct training and 2) finetuning scenarios with volumetric multi-organ/tissues segmentation. 3D UX-Net achieves consistently improvement in both scenarios across all ConvNets and transformers SOTA with fewer model parameters.

## 2 RELATED WORK

### 2.1 TRANSFORMER-BASED SEGMENTATION

Significant efforts have been put into integrating ViTs for dense predictions in medical imaging domain Hatamizadeh et al. (2022b); Chen et al. (2021); Zhou et al. (2021); Wang et al. (2021). With the advancement of Swin Transformer, SwinUNETR equips the encoder with the Swin Transformer blocks to compute self-attention for enhancing brain tumor segmentation accuracy in 3D MRI Images Hatamizadeh et al. (2022a). Tang et al. extends the SwinUNETR by adding a self-supervised learning pre-training strategy for fine-tuning segmentation tasks. Another Unet-like architecture Swin-Unet further adapts Swin Transformer on both the encoder and decoder network via skip-connections to learn local and global semantic features for multi-abdominal CT segmentation Cao et al. (2021). Similarly, SwinBTS has the similar intrinsic structure with Swin-Unet with an enhanced transformer module for detailed feature extraction Jiang et al. (2022). However, the

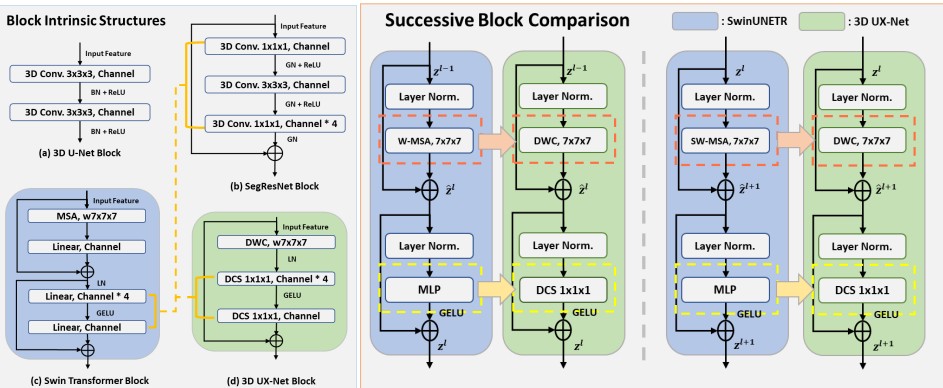

Figure 1: Overview of our proposed designed convolution blocks to simulate the behaviour of swin transformers. We leverage depthwise convolution and pointwise scaling to adapt large receptive field and enrich the features through widening independent channels. We further compare different backbones of volumetric ConvNets and Swin Transformer block architecture. The yellow dotted line demonstrates the differences in spatial position of widening feature channels in the network bottleneck.

transformer-based volumetric segmentation frameworks still require lengthy training time and are accompanied by high computational complexity associated with extracting features at multi-scale levels Xie et al. (2021); Shamshad et al. (2022). Therefore, such limitations motivate us to rethink if ConvNets can emulate transformer behavior to demonstrate efficient feature extraction.

## 2.2 DEPTHWISE CONVOLUTION BASED SEGMENTATION

Apart from transformer-based framework, previous works began to revisit the concept of depthwise convolution and adapt its characteristics for robust segmentation. It has been proved to be a powerful variation of standard convolution that helps reduce the number of parameters and transfer learning Guo et al. (2019). Zunair et al. introduced depthwise convolution to sharpen the features prior to fuse the decode features in a UNet-like architecture Zunair & Hamza (2021). 3D $U^2$-Net leveraged depthwise convolutions as domain adaptors to extract domain-specific features in each channel Huang et al. (2019). Both studies demonstrate the feasibility of using depthwise convolution in enhancing volumetric tasks. However, only a small kernel size is used to perform depthwise convolution. Several prior works have investigated the effectiveness of LK convolution in medical image segmentation. For instance, Huo et al. leveraged LK (7x7) convolutional layers as the skip connections to address the anatomical variations for splenomegaly spleen segmentation Huo et al. (2018); Li et al. proposed to adapt LK and dilated depthwise convolutions in decoder for volumetric segmentation Li et al. (2022). However, significant increase of FLOPs is demonstrated with LKs and dramatically reduces both training and inference efficiency. To enhance the model efficiency with LKs, Liu et al. proposed ConvNeXt as a 2D generic backbone that simulate ViTs advantages with LK depthwise convolution for downstream tasks with natural image Liu et al. (2022), while ConvUNeXt is proposed to extend for 2D medical image segmentation and compared only with 2D CNN-based networks (e.g., ResUNet Shu et al. (2021), UNet++ Zhou et al. (2019)) Han et al. (2022). However, limited studies have been proposed to efficiently leverage depthwise convolution with LKs in a volumetric setting and compete favorably with volumetric transformer approaches. With the large receptive field brought by LK depthwise convolution, we hypothesize that LK depthwise convolution can potentially emulate transformers' behavior and efficiently benefits for volumetric segmentation.

## 3 3D UX-NET: INTUITION

Inspired by Liu et al. (2022), we introduce 3D UX-Net, a simple volumetric ConvNet that adapts the capability of hierarchical transformers and preserves the advantages of using ConvNet modules such as inductive biases. The basic idea of designing the encoder block in 3D UX-Net can be divided into

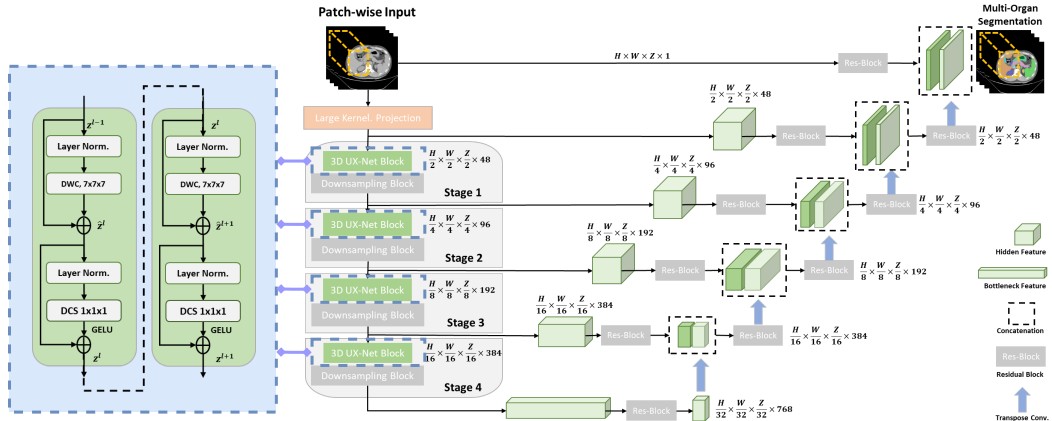

Figure 2: Overview of the proposed 3D UX-Net with our designed convolutional block as the encoder backbone. LK convolution is used to project features into patch-wise embeddings. A downsampling block is used in each stage to mix and enrich context across all channels, while our designed blocks extract meaningful features in depth-wise setting.

1) block-wise and 2) layer-wise perspectives. First, we discuss the block-wise perspective in three views:

- **Patch-wise Features Projection**: Comparing the similarities between ConvNets and ViTs, there is a common block that both networks use to aggressively downscale feature representations into particular patch sizes. Here, instead of flattening image patches as a sequential input with linear layer Dosovitskiy et al. (2020), we adopt a LK projection layer to extract patch-wise features as the encoder's inputs.

- **Volumetric Depth-wise Convolution with LKs**: One of the intrinsic specialties of the swin transformer is the sliding window strategy for computing non-local MSA. Overall, there are two hierarchical ways to compute MSA: 1) window-based MSA (W-MSA) and 2) shifted window MSA (SW-MSA). Both ways generate global receptive field across layers and further refine the feature correspondence between non-overlapping windows. Inspired by the idea of depth-wise convolution, we have found similarities between the weighted sum approach in self-attention and the convolution per-channel basis. We argue that using depth-wise convolution with a LK size can provide a large receptive field in extracting features similar to the MSA blocks. Therefore, we propose compressing the window shifting characteristics of the Swin Transformer with a volumetric depth-wise convolution using a LK size (e.g., starting from $7 \times 7 \times 7$). Each kernel channel is convolved with the corresponding input channel, so that the output feature has the same channel dimension as the input.

- **Inverted Bottleneck with Depthwise Convolutional Scaling**: Another intrinsic structure in Swin Transformer is that they are designed with the hidden dimension of the MLP block to be four times wider than the input dimension, as shown in Figure 1. Such a design is interestingly correlated to the expansion ratio in the ResNet block He et al. (2016). Therefore, we leverage the similar design in ResNet block and move up the depth-wise convolution to compute features. Furthermore, we introduce depthwise convolutional scaling (DCS) with $1 \times 1 \times 1$ kernel to linearly scale each channel feature independently. We enrich the feature representations by expanding and compressing each channel independently, thus minimizing the redundancy of cross-channel context. We enhance the cross-channel feature correspondences with the downsampling block in each stage. By using DCS, we further reduce the model complexity by 5% and demonstrates a comparable results with the block architecture using MLP.

The macro-design in convolution blocks demonstrates the possibility of adapting the large receptive field and leveraging similar operation of extracting features compared with the Swin Transformer. We want to further investigate the variation between ConvNets and the Swin Transformer in layer-

wise settings and refine the model architecture to better simulate ViTs in macro-level. Here, we further define and adapt layer-wise differences into another three perspectives:

- **Applying Residual Connections**: From Figure 1, the golden standard 3D U-Net block demonstrates the naive approach of using small kernels to extract local representations with increased channels Çiçek et al. (2016), while the SegResNet block applies the residual similar to the transformer block Myronenko (2018). Here, we also apply residual connections between the input and the extracted features after the last scaling layer. However, we do not apply any normalization and activation layers before and after the summation of residual to be equivalent with the swin transformer structure.

- **Adapting Layer Normalization (LN)**: In ConvNets, batch normalization (BN) is a common strategy that normalizes convolved representations to enhance convergence and reduce overfitting. However, previous works have demonstrated that BN can lead to a detrimental effect in model generalizability Wu & Johnson (2021). Although several approaches have been proposed to have an alternative normalization techniques Salimans & Kingma (2016); Ulyanov et al. (2016); Wu & He (2018), BN still remains as the optimal choice in volumetric vision tasks. Motivated by vision transformers and Liu et al. (2022), we directly substitute BN with LN in the encoder block and demonstrate similar operations in ViTs Ba et al. (2016).

- **Using GELU as the Activation Layer**: Many previous works have used the rectified linear unit (ReLU) activation layers Nair & Hinton (2010), providing non-linearity in both ConvNets and ViTs. However, previously proposed transformer models demonstrate the Gaussian error linear unit (GELU) to be a smoother variant, which tackle the limitation of sudden zero in the negative input range in ReLU Hendrycks & Gimpel (2016). Therefore, we further substitute the ReLU with the GELU activation function.

## 4   3D UX-NET: COMPLETE NETWORK DESCRIPTION

3D UX-Net comprises multiple re-designed volumetric convolution blocks that directly utilize 3D patches. Skip connections are further leveraged to connect the multi-resolution features to a convolution-based decoder network. Figure 2 illustrates the complete architecture of 3D UX-Net. We further describe the details of the encoder and decoder in this section.

### 4.1   DEPTH-WISE CONVOLUTION ENCODER

Given a set of 3D image volumes $V_i = X_i, Y_{i_{i=1,...,L}}$, random sub-volumes $P_i \in \mathcal{R}^{H \times W \times D \times C}$ are extracted to be the inputs for the encoder network. Instead of flattening the patches and projecting it with linear layer Hatamizadeh et al. (2022b), we leverage a LK convolutional layer to compute partitioned feature map with size $\frac{H}{2} \times \frac{W}{2} \times \frac{D}{2}$ that are projected into a $C = 48$-dimensional space. To adapt the characteristics of computing local self-attention, we use the depthwise convolution with kernel size starting from $7 \times 7 \times 7$ (DWC) with padding of 3, to act as a "shifted window" and evenly divide the feature map. As global self-attention is generally not computationally affordable with a large number of patches extracted in the Swin Transformer Liu et al. (2021), we hypothesize that performing depthwise convolution with a LK size can effectively extract features with a global receptive field. Therefore, we define the output of encoder blocks in layers $l$ and $l + 1$ as follows:

$$
\begin{aligned}
\hat{z}^l &= \text{DWC}(\text{LN}(z^{l-1})) + z^{l-1} \\
z^l &= \text{DCS}(\text{LN}(\hat{z}^l)) + \hat{z}^l \\
\hat{z}^{l+1} &= \text{DWC}(\text{LN}(z^l)) + z^l \\
z^{l+1} &= \text{DCS}(\text{LN}(\hat{z}^{l+1})) + \hat{z}^{l+1}
\end{aligned}
\tag{1}
$$

where $\hat{z}_l$ and $\hat{z}_{l+1}$ are the outputs from the DWC layer in different depth levels; LN and DCS denote as the layer normalization and the depthwise convolution scaling, respectively (see. Figure 1). Compared to the Swin Transformer, we substitute the regular and window partitioning multi-head self-attention modules, W-MSA and SW-MSA respectively, with two DWC layers.

Motivated by SwinUNETR Tang et al. (2022); Hatamizadeh et al. (2022a), the complete architecture of the encoder consists of 4 stages comprising of 2 LK convolution blocks at each stage (*i.e.* L=8

Table 1: Comparison of transformer and ConvNet SOTA approaches on the Feta 2021 and FLARE 2021 testing dataset. (*: $p < 0.01$, with Wilcoxon signed-rank test to all SOTA approaches)

| Methods | #Params | FLOPs | FeTA 2021 | | | | | | | | FLARE 2021 | | | | |
|---|---|---|---|---|---|---|---|---|---|---|---|---|---|---|---|
| | | | ECF | GM | WM | Vent. | Cereb. | DGM | BS | Mean | Spleen | Kidney | Liver | Pancreas | Mean |
| 3D U-Net Çiçek et al. (2016) | 4.81M | 135.9G | 0.867 | 0.762 | 0.925 | 0.861 | 0.910 | 0.845 | 0.827 | 0.857 | 0.911 | 0.962 | 0.905 | 0.789 | 0.892 |
| SegResNet Myronenko (2018) | 1.18M | 15.6G | 0.868 | 0.770 | 0.927 | 0.865 | 0.911 | 0.867 | 0.825 | 0.862 | 0.963 | 0.934 | 0.965 | 0.745 | 0.902 |
| RAP-Net Lee et al. (2021) | 38.2M | 101.2G | 0.880 | 0.771 | 0.927 | 0.862 | 0.907 | 0.879 | 0.832 | 0.865 | 0.946 | 0.967 | 0.940 | 0.799 | 0.913 |
| nn-UNet Isensee et al. (2021) | 31.2M | 743.3G | 0.883 | 0.775 | 0.930 | 0.868 | **0.920** | 0.880 | 0.840 | 0.870 | 0.971 | 0.966 | 0.976 | 0.792 | 0.926 |
| TransBTS Wang et al. (2021) | 31.6M | 110.4G | **0.885** | 0.778 | 0.932 | 0.861 | 0.913 | 0.876 | 0.837 | 0.868 | 0.964 | 0.959 | 0.974 | 0.711 | 0.902 |
| UNETR Hatamizadeh et al. (2022b) | 92.8M | 82.6G | 0.861 | 0.762 | 0.927 | 0.862 | 0.908 | 0.868 | 0.834 | 0.860 | 0.927 | 0.947 | 0.960 | 0.710 | 0.886 |
| nnFormer Zhou et al. (2021) | 149.3M | 240.2G | 0.880 | 0.770 | 0.930 | 0.857 | 0.903 | 0.876 | 0.828 | 0.863 | 0.973 | 0.960 | 0.975 | 0.717 | 0.906 |
| SwinUNETR Hatamizadeh et al. (2022a) | 62.2M | 328.4G | 0.873 | 0.772 | 0.929 | 0.869 | 0.914 | 0.875 | 0.842 | 0.867 | 0.979 | 0.965 | 0.980 | 0.788 | 0.929 |
| **3D UX-Net (Ours)** | 53.0M | 639.4G | 0.882 | **0.780** | **0.934** | **0.872** | 0.917 | **0.886** | **0.845** | **0.874\*** | **0.981** | **0.969** | **0.982** | **0.801** | **0.934\*** |

Table 2: Comparison of Finetuning performance with transformer SOTA approaches on the AMOS 2021 testing dataset.(*: $p < 0.01$, with Wilcoxon signed-rank test to all SOTA approaches)

| Methods | Spleen | R. Kid | L. Kid | Gall. | Eso. | Liver | Stom. | Aorta | IVC | Panc. | RAG | LAG | Duo. | Blad. | Pros. | Avg |
|---|---|---|---|---|---|---|---|---|---|---|---|---|---|---|---|---|
| nn-UNet | 0.965 | 0.959 | 0.951 | 0.889 | 0.820 | 0.980 | 0.890 | 0.948 | 0.901 | 0.821 | 0.785 | 0.739 | 0.806 | 0.869 | 0.839 | 0.878 |
| TransBTS | 0.885 | 0.931 | 0.916 | 0.817 | 0.744 | 0.969 | 0.837 | 0.914 | 0.855 | 0.724 | 0.630 | 0.566 | 0.704 | 0.741 | 0.650 | 0.792 |
| UNETR | 0.926 | 0.936 | 0.918 | 0.785 | 0.702 | 0.969 | 0.788 | 0.893 | 0.828 | 0.732 | 0.717 | 0.554 | 0.658 | 0.683 | 0.722 | 0.762 |
| nnFormer | 0.935 | 0.904 | 0.887 | 0.836 | 0.712 | 0.964 | 0.798 | 0.901 | 0.821 | 0.734 | 0.665 | 0.587 | 0.641 | 0.744 | 0.714 | 0.790 |
| SwinUNETR | 0.959 | 0.960 | 0.949 | 0.894 | 0.827 | 0.979 | 0.899 | 0.944 | 0.899 | 0.828 | 0.791 | **0.745** | 0.817 | 0.875 | 0.841 | 0.880 |
| 3D UX-Net | **0.970** | **0.967** | **0.961** | **0.923** | **0.832** | **0.984** | **0.920** | **0.951** | **0.914** | **0.856** | **0.825** | 0.739 | **0.853** | **0.906** | **0.876** | **0.900\*** |

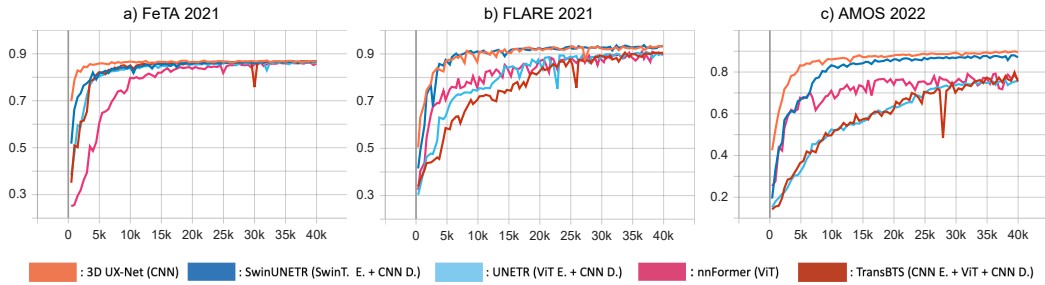

Figure 3: Validation Curve with Dice Score for FeTA2021 (a), FLARE2021 (b) and AMOS2022 (c). 3D UX-Net demonstrates the fastest convergence rate with limited samples training (FeTA2021) and transfer learning (AMOS2022) scenario respectively, while the convergence rate is comparable to SwinUNETR with the increase of sample size training (FLARE2021).

total layers). Inside the block, the DCS layer is followed by the DWC layer in each block. The DCS layer helps scale the dimension of the feature map (4 times of the input channel size) without increasing model parameters and minimizes the redundancy of the learned volumetric context across channels. To exchange the information across channels, instead of using MLP, we leverage a standard convolution block with kernel size $2 \times 2 \times 2$ with stride 2 to downscale the feature resolution by a factor of 2. The same procedure continues in stage 2, stage 3 and stage 4 with the resolutions of $\frac{H}{4} \times \frac{W}{4} \times \frac{D}{4}$, $\frac{H}{8} \times \frac{W}{8} \times \frac{D}{8}$ and $\frac{H}{16} \times \frac{W}{16} \times \frac{D}{16}$ respectively. Such hierarchical representations in multi-scale setting are extracted in each stage and are further leveraged for learning dense volumetric segmentation.

## 4.2 DECODER

The multi-scale output from each stage in the encoder is connected to a ConvNet-based decoder via skip connections and form a "U-shaped" like network for downstream segmentation task. Specifically, we extract the output feature mapping of each stage $i(i \in 0, 1, 2, 3, 4)$ in the encoder and further leverage a residual block comprising two post-normalized $3 \times 3 \times 3$ convolutional layers with instance normalization to stabilize the extracted features. The processed features from each stage are then upsampled with a transpose convolutional layer and concatentated with the features from the preceding stage. For downstream volumetric segmentation, we also concatenate the residual features from the input patches with the upsampled features and input the features into a residual

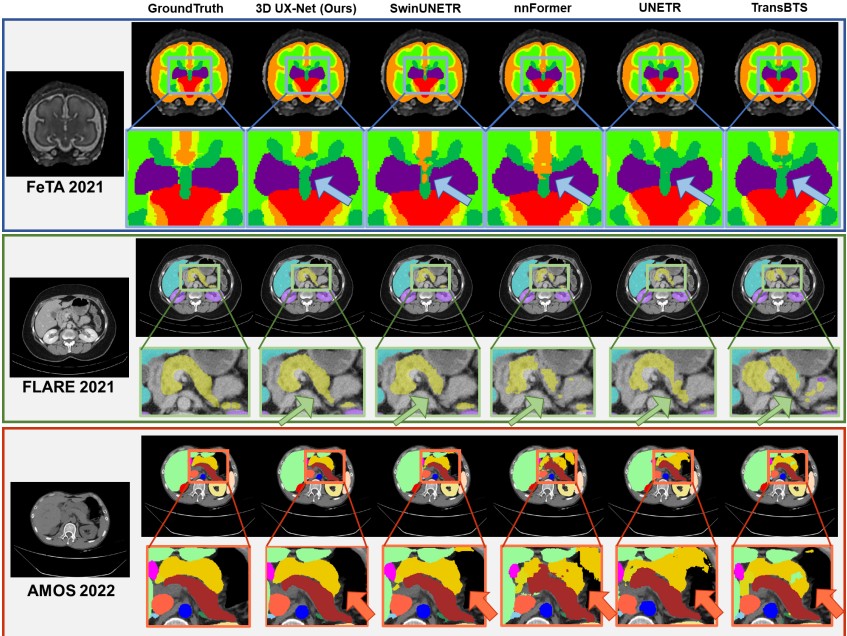

Figure 4: Qualitative representations of tissues and multi-organ segmentation across three public datasets. Boxed are further zoomed in and visualize the significant differences in segmentation quality. 3D UX-Net shows the best segmentation quality compared to the ground-truth.

block with $1 \times 1 \times 1$ convolutional layer with a softmax activation to predict the segmentation probabilities.

## 5 EXPERIMENTAL SETUP

**Datasets** We conduct experiments on three public multi-modality datasets for volumetric segmentation, which comprising with 1) MICCAI 2021 FeTA Challenge dataset (FeTA2021) Payette et al. (2021), 2) MICCAI 2021 FLARE Challenge dataset (FLARE2021) Ma et al. (2021), and 3) MICCAI 2022 AMOS Challenge dataset (AMOS2022) Ji et al. (2022). For the FETA2021 dataset, we employ 80 T2-weighted infant brain MRIs from the University Children's Hospital with 1.5 T and 3T clinical whole-body scanners for brain tissue segmentation, with seven specific tissues well-annotated. For FLARE2021 and AMOS2022, we employ 511 multi-contrast abdominal CT from FLARE2021 with four anatomies manually annotated and 200 multi-contrast abdominal CT from AMOS 2022 with sixteen anatomies manually annotated for abdominal multi-organ segmentation. More details of the three public datasets can be found in appendix A.2.

**Implementation Details** We perform evaluations on two scenarios: 1) direct supervised training and 2) transfer learning with pretrained weights. FeTA2021 and FLARE2021 datasets are leverage to evaulate in direct training scenario, while AMOS dataset is used in transfer learning scenario. We perform five-fold cross-validations to both FeTA2021 and FLARE2021 datasets. More detailed information of data splits are provided in Appendix A.2. For the transfer learning scenario, we leverage the pretrained weights from the best fold model trained with FLARE2021, and finetune the model weights on AMOS2022 to evaluate the fine-tuning capability of 3D UX-Net. The complete preprocessing and training details are available at the appendix A.1. Overall, we evaluate 3D UX-Net performance by comparing with current volumetric transformer and ConvNet SOTA approaches for volumetric segmentation in fully-supervised setting. We use the Dice similarity coefficient as an evaluation metric to compare the overlapping regions between predictions and ground-truth labels. Furthermore, we performed ablation studies to investigate the effect on different kernel size and the variability of substituting linear layers with depthwise convolution for feature extraction.

Table 3: Ablation studies of different architecture on FeTA2021 and FLARE2021

| Methods | #Params (M) | FeTA2021 | FLARE2021 |
|---|---|---|---|
| | | Mean Dice | |
| SwinUNETR | 62.2 | 0.867 | 0.929 |
| Use Standard Conv. | 186.9 | 0.875 | 0.937 |
| Use Depth Conv. | 53.0 | 0.874 | 0.934 |
| Kernel=$3 \times 3 \times 3$ | 52.5 | 0.867 | 0.928 |
| Kernel=$5 \times 5 \times 5$ | 52.7 | 0.869 | 0.931 |
| Kernel=$7 \times 7 \times 7$ | 53.0 | 0.874 | 0.934 |
| Kernel=$9 \times 9 \times 9$ | 53.6 | 0.870 | 0.934 |
| Kernel=$11 \times 11 \times 11$ | 54.4 | 0.871 | 0.936 |
| Kernel=$13 \times 13 \times 13$ | 55.7 | 0.871 | 0.938 |
| No MLP | 51.1 | 0.869 | 0.915 |
| Use MLP | 56.3 | 0.872 | 0.933 |
| Use DCS $1 \times 1 \times 1$ | 53.0 | 0.874 | 0.934 |

## 6 RESULTS

### 6.1 EVALUATION ON FeTA & FLARE

Table 1 shows the result comparison of current transformers and ConvNets SOTA on medical image segmentation in volumetric setting. With our designed convolutional blocks as the encoder backbone, 3D UX-Net demonstrates the best performance across all segmentation task with significant improvement in Dice score (FeTA2021: 0.870 to 0.874, FLARE2021: 0.929 to 0.934). From Figure 2, we observe that 3D UX-Net demonstrates the quickest convergence rate in training with FeTA2021 datasets. Interestingly, when the training sample size increases, the efficiency of training convergence starts to become compatible between SwinUNETR and 3D UX-Net. Apart from the quantitative representations, Figure 3 further provides additional confidence of demonstrating the quality improvement in segmentation with 3D UX-Net. The morphology of organs and tissues are well preserved compared to the ground-truth label.

### 6.2 TRANSFER LEARNING WITH AMOS

Apart from training from scratch scenario, we further investigate the transfer learning capability of 3D UX-Net comparing to the transformers SOTA with AMOS 2022 dataset. We observe that the finetuning performance of 3D UX-Net significantly outperforms other transformer network with mean Dice of 0.900 (2.27% enhancement) and most of the organs segmentation demonstrate a consistent improvement in quality. Also, from Figure 2, although the convergence curve of each transformer network shows the comparability to that of the FLARE2021-trained model, 3D UX-Net further shows its capability in adapting fast convergence and enhancing the robustness of the model with finetuning. Furthermore, the qualitative representations in Figure 3 demonstrates a significant improvement in preserving boundaries between neighboring organs and minimize the possibility of over-segmentation towards other organ regions.

### 6.3 ABLATION ANALYSIS

After evaluating the core performance of 3D UX-Net, we study how the different components in our designed architecture contribute to such a significant improvement in performance, as well as how they interact with other components. Here, both FeTA2021 and FLARE2021 are leveraged to perform ablation studies towards different modules. All ablation studies are performed with kernel size $7 \times 7 \times 7$ scenario except the study of evaluating the variability of kernel size.

**Comparing with Standard Convolution:** We investigate the effectiveness of both standard convolution and depthwise convolution for initial feature extraction. With the use of standard convolution, it demonstrates a slight improvement with standard convolution. However, the model parameters are about 3.5 times than that of using depthwise convolution, while the segmentation performance

with depthwise convolution still demonstrates a comparable performance in both datasets.

**Variation of Kernel Size:** From Table 3, we observe that the convolution with kernel size $7 \times 7 \times 7$ optimally works for FeTA2021 dataset, while the segmentation performance of FLARE2021 demonstrates the best with kernel size of $13 \times 13 \times 13$. The significant improvement of using $13 \times 13 \times 13$ kernel for FLARE2021 may be due to the larger receptive field provided to enhance the feature correspondence between multiple neighboring organs within the abdominal region. For FeTA2021 dataset, only the small infant brains are well localized as foreground and $7 \times 7 \times 7$ kernel demonstrates to be optimal recpetive field to extract the tissues correspondence.

**Adapting DCS:** We found that a significant decrement is performed without using MLP for feature scaling. With the linear scaling, the performance enhanced significantly in FLARE2021, while a slight improvement is demonstrated in FeTA2021. Interestingly, leveraging depthwise convolution with $1 \times 1 \times 1$ kernel size for scaling, demonstrates a slightly enhancement in performance for both FeTA2021 and FLARE2021 datasets. Also, the model parameters further drops from 56.3M to 53.0M without trading off the model performance.

## 7    DISCUSSION

In this work, we present a block-wise design to simulate the behavior of Swin Transformer using pure ConvNet modules. We further adapt our design as a generic encoder backbone into "U-Net" like architecture via skip connections for volumetric segmentation. We found that the key components for improved performance can be divided into two main perspectives: 1) the sliding window strategy of computing MSA and 2) the inverted bottleneck architecture of widening the computed feature channels. The W-MSA enhances learning the feature correspondence within each window, while the SW-MSA strengthens the cross-window connections at the feature level between different non-overlapping windows. Such strategy integrates ConvNet priors into transformer networks and enlarge receptive fields for feature extraction. However, we found that the depth convolutions can demonstrate similar operations of computing MSA in Swin Transformer blocks. In depth-wise convolutions, we convolve each input channel with a single convolutional filter and stack the convolved outputs together, which is comparable to the patch merging layer for feature outputs in swin transformers. Furthermore, adapting the depth convolutions with LK filters demonstrates similarities with both W-MSA and SW-MSA, which learns the feature connections within a large receptive field. Our design provides similar capabilities to Swin Transformer and additionally has the advantage of reducing the number of model parameters using ConvNet modules.

Another interesting difference is the inverted bottleneck architecture. Figure 1 shows that both Swin Transformer and some standard ConvNets have their specific bottleneck architectures (yellow dotted line). The distinctive component in swin transformer's bottleneck is to maintain the channels size as four times wider than the input dimension and the spatial position of the MSA layer. We follow the inverted bottleneck architecture in Swin Transformer block and move the depthwise convolution to the top similar to the MSA layer. Instead of using linear scaling, we introduce the idea of depthwise convolution in pointwise setting to scale the dense feature with wider channels. Interestingly, we found a slight improvement in performance is shown across datasets (FeTA2021: 0.872 to 0.874, FLARE2021: 0.933 to 0.934), but with less model parameters. As each encoder block only consists of two scaling layers, the limited number of scaling blocks may affect the performance to a small extent. We will further investigate the scalability of linear scaling layer in 3D as the future work.

## 8    CONCLUSION

We introduce 3D UX-Net, the first volumetric network adapting the capabilities of hierarchical transformer with pure ConvNet modules for medical image segmentation. We re-design the encoder blocks with depthwise convolution and projections to simulate the behavior of hierarchical transformer. Furthermore, we adjust layer-wise design in the encoder block and enhance the segmentation performance across different training settings. 3D UX-Net outperforms current transformer SOTAs with fewer model parameters using three challenging public datasets in both supervised training and transfer learning scenarios.

ACKNOWLEDGMENTS

This research is supported by NIH Common Fund and National Institute of Diabetes, Digestive and Kidney Diseases U54DK120058 (Spraggins), NSF CAREER 1452485, NIH 2R01EB006136, NIH 1R01EB017230 (Landman), and NIH R01NS09529. This study was in part using the resources of the Advanced Computing Center for Research and Education (ACCRE) at Vanderbilt University, Nashville, TN. The identified datasets used for the analysis described were obtained from the Research Derivative (RD), database of clinical and related data. The imaging dataset(s) used for the analysis described were obtained from ImageVU, a research repository of medical imaging data and image-related metadata. ImageVU and RD are supported by the VICTR CTSA award (ULTR000445 from NCATS/NIH) and Vanderbilt University Medical Center institutional funding. ImageVU pilot work was also funded by PCORI (contract CDRN-1306-04869). We further thank Quan Liu, a Ph.D student in Computer Science Department of Vanderbilt University, to extensively discuss the initial idea of this paper.

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

# A  APPENDIX

## A.1  DATA PREPROCESSING & MODEL TRAINING

We apply hierarchical steps for data preprocessing: 1) intensity clipping is applied to further enhance the contrast of soft tissue (FLARE2021 & AMOS2022:{min:-175, max:250}). 2) Intensity normalization is performed after clipping for each volume and use min-max normalization: $(X - X_1)/(X_{99} - X_1)$ to normalize the intensity value between 0 and 1, where $X_p$ denote as the $p_{th}$ percentile of intensity in $X$. We then randomly crop sub-volumes with size $96 \times 96 \times 96$ at the foreground and perform data augmentations, including rotations, intensity shifting, and scaling (scaling factor: 0.1). All training processes with 3D UX-Net are optimized with an AdamW optimizer. We trained all models for 40000 steps using a learning rate of 0.0001 on an NVIDIA-Quadro RTX 5000 for both FeTA2021 and FLARE2021, while we perform training for AMOS2022 using NVIDIA-Quadro RTX A6000. One epoch takes approximately about 1 minute for FeTA2021, 10 minutes for FLARE2021, and 7 minutes for AMOS2022, respectively. We further summarize all the training parameters with Table 4.

Table 4: Hyperparameters of both directly training and finetuning scenarios on three public datasets

| Hyperparameters | Direct Training | Finetuning |
|---|---|---|
| Encoder Stage | 4 | |
| Layer-wise Channel | $48, 96, 192, 384$ | |
| Hidden Dimensions | 768 | |
| Patch Size | $96 \times 96 \times 96$ | |
| No. of Sub-volumes Cropped | 2 | 1 |
| Training Steps | 40000 | |
| Batch Size | 2 | 1 |
| AdamW $\epsilon$ | $1e - 8$ | |
| AdamW $\beta$ | $(0.9, 0.999)$ | |
| Peak Learning Rate | $1e - 4$ | |
| Learning Rate Scheduler | ReduceLROnPlateau | N/A |
| Factor & Patience | 0.9, 10 | N/A |
| Dropout | X | |
| Weight Decay | 0.08 | |
| Data Augmentation | Intensity Shift, Rotation, Scaling | |
| Cropped Foreground | ✓ | |
| Intensity Offset | 0.1 | |
| Rotation Degree | $-30°$ to $+30°$ | |
| Scaling Factor | x: 0.1, y: 0.1, z: 0.1 | |

## A.2  PUBLIC DATASETS DETAILS

Table 5: Complete Overview of three public MICCAI Chanllenge Datasets

| MICCAI Challenge | FeTA 2021 | FLARE 2021 | AMOS 2022 |
|---|---|---|---|
| Imaging Modality | 1.5T & 3T MRI | Multi-Contrast CT | Multi-Contrast CT |
| Anatomical Region | Infant Brain | Abdomen | Abdomen |
| Dimensions | $256 \times 256 \times 256$ | $512 \times 512 \times \{37 - 751\}$ | $512 - 768 \times 512 - 768 \times \{68 - 353\}$ |
| Resolution | $\{0.43 - 0.70\} \times \{0.43 - 0.70\} \times \{0.43 - 0.70\}$ | $\{0.61 - 0.98\} \times \{0.61 - 0.98\} \times \{0.50 - 7.50\}$ | $\{0.45 - 1.07\} \times \{0.45 - 1.07\} \times \{1.25 - 5.00\}$ |
| Sample Size | 80 | 361 | 200 |
| Anatomical Label | External Cerebrospinal Fluid (ESF), Grey Matter (GM), White Matter (WM), Ventricles, Cerebellum, Deep Grey Matter (DGM) Brainstem | Spleen, Kidney, Liver, Pancreas | Spleen, Left & Right Kidney, Gall Bladder, Esophagus, Liver, Stomach, Aorta, Inferior Vena Cava (IVC) Pancreas, Left & Right Adrenal Gland (AG), Duodenum, Bladder, Prostates/uterus |
| Data Splits | 5-Fold Cross-Validation Train: 50 / Validation: 12 / Test: 18 | 5-Fold Cross-Validation Train: 272 / Validation: 69 / Test: 20 | 1-Fold Train: 160 / Validation: 20 / Test: 20 |

## A.3  FURTHER DISCUSSIONS COMPARING TO NN-UNET

In Table 1 & 2, we compare our proposed network with multiple CNN-based SOTA networks and the golden standard approach nn-UNet. We observe that the performance of nn-UNet nearly outperform most of the transformer state-of-the-arts in both FeTA 2021 and FLARE 2021 datasets. Such

Table 6: Albation Studies of Adapting nn-UNet architecture on the Feta 2021 and FLARE 2021 testing dataset. (*: $p < 0.01$, with Wilcoxon signed-rank test to all SOTA approaches, D.S: Deep Supervision)

| | FeTA 2021 | | | | | | | | FLARE 2021 | | | | |
|---|---|---|---|---|---|---|---|---|---|---|---|---|---|
| Methods | ECF | GM | WM | Vent. | Cereb. | DGM | BS | Mean | Spleen | Kidney | Liver | Pancreas | Mean |
| nn-UNet Isensee et al. (2021) | 0.883 | 0.775 | 0.930 | 0.868 | 0.920 | 0.880 | 0.840 | 0.870 | 0.971 | 0.966 | 0.976 | 0.792 | 0.926 |
| **3D UX-Net (Plain)** | 0.882 | 0.780 | 0.934 | 0.872 | 0.917 | 0.886 | 0.845 | 0.874 | 0.981 | 0.969 | 0.982 | 0.801 | 0.934 |
| **3D UX-Net (nn-UNet struct., w/o D.S.)** | 0.885 | 0.784 | 0.937 | 0.872 | 0.921 | 0.887 | 0.849 | 0.876 | **0.983** | **0.972** | **0.983** | 0.821 | **0.940***  |
| **3D UX-Net (nn-UNet struct., D.S.)** | **0.890** | **0.791** | **0.939** | **0.877** | **0.922** | **0.891** | **0.854** | **0.881*** | **0.986** | **0.974** | **0.983** | **0.833** | **0.944*** |

improvement may mainly contribute to its innovation of self-configuration training strategies and en-sembling outputs as postprocessing technique, while the network used in nn-UNet is only the plain 3D U-Net architecture. To further characterize the ability of our proposed network, we further sub-stitute the plain 3D U-Net architecture with our proposed 3D UX-Net and adapt the self-configuring hyperparameters for training. We demonstrate a significant improvement of performance in FeTA 2021 and FLARE 2021 datasets with mean organ Dice from 0.874 to 0.881 and from 0.934 to 0.944 respectively, as shown in Table 6. To further investigate the difference in the network architecture, we observed that the convolution blocks in nn-UNet leverage the combination of instance normaliza-tion and leakyReLU. Such design allows to normalize channel-wise feature independently and mix the channel context with small kernel convolutional layers. In our design, we provide an alternative thought of extracting channel-wise features independently with depthwise convolution and mix the channel information during the downsampling layer only. Therefore, layer normalization is lever-aged in our scenario and we want to further enhance the feature correspondence with large receptive field across channels efficiently. Furthermore, we found that the deep supervision strategy in nn-UNet, which compute an auxilary loss with each stages' intermediate output, also demonstrates its effectiveness to further improve the performance (FeTA 2021: from 0.876 to 0.881; FLARE 2021: from 0.940 to 0.944).

For the training scenarios, instead of using the proposed initial learning rate 0.01, we reduce the initial learning rate to 0.002 to train with 150 epochs (40000 steps $\approx$ 150 epochs) for FLARE 2021 and 850 epochs (40000 steps $\approx$ 850 epochs) for FeTA 2021 respectively, with the batch size of 2. For the finetuning scenario with AMOS 2022, we only train the nn-UNet model with 250 epochs (40000 steps $\approx$ 250 epochs), instead of the default settings (1000 epochs) to ensure the fair network comparison with similar steps.

## A.4 Further Discussions on Training and Inference Efficiency

Table 7: Albation Studies of Optimizing 3D U-XNet architecture on the Feta 2021 and FLARE 2021 testing dataset. (SD: Stage Depth, HDim: Hidden Dimension in the Bottleneck Layer.)

| Methods | #Params (M) | FLOPs (G) | FeTA2021 | FLARE2021 |
|---|---|---|---|---|
| | | | Mean Dice | |
| nn-UNet | 31.2M | 743.3G | 0.870 | 0.926 |
| SwinUNETR | 62.2M | 328.4G | 0.867 | 0.929 |
| SD: 2,2,2,2, HDim: 768 | 53.0M | 639.4G | 0.874 | 0.934 |
| SD: 2,2,8,2, HDim: 384 | 32.1M | 536.8G | 0.873 | 0.932 |

Apart from the advantage of quantitative performance, we further leverage the LK depthwise con-volutions to reduce the model parameters from 62.2M to 53.0M, compared to SwinUNETR in Table 3. However, although the training efficiency of 3D UX-Net is already better than nn-UNet (FLOPs: 743.3G to 639.4G), we observed that the FLOPs of 3D UX-Net still remains at a high value. In-spired by the architectures of both Swin Transformer Liu et al. (2021) and ConvNeXt Liu et al. (2022) used in the natural image domain, we further remove the bottleneck layer (ResNet block with 768 channels) and increase the block depth of stage 3 (e.g., 8 blocks). Such optimized design further significantly reduces both the model parameters (from 53.0M to 32.1M, nn-UNet: 31.2M) and FLOPs (from 639.4G to 536.1G, nn-UNet: 743.3M), while preserving the performance (shown

in Table 7). Additional validation studies is needed to investigate the effectiveness of both MLP and pointwise DCS, and optimizing 3D UX-Net architecture, which will be the next steps of our future work. Another observation in Table 3 is the subtle differences in model parameters between kernel size of $3 \times 3 \times 3$ and $7 \times 7 \times 7$. We found that the increase of both model parameters and FLOPs is also attributed to the design of decoder network. Our decoder block design further add a 3D ResNet block after the transpose convolution to further resample and mix the channel context, instead of directly perform transpose convolution in nn-UNet. A efficient block design in decoder network is demanded to be further investigated and using depthwise convolution may be another potential solution to reduce the low efficiency burden.

To further reduce the burden of low training and inference efficiency, re-parameterization of LK convolutional blocks may be another promising direction to focus. Prior works have demonstrated to scale up few convolutional blocks with LK size $(31 \times 31)$ and propose the idea of parallel branches with small kernels for residual shortcuts Ding et al. (2022b; 2021; 2022a). The parallel branch can then be mutually converted through equivalent transformation of parameters. For example. a branch of $1 \times 1$ convolution and a branch of $7 \times 7$ convolution, can be transferred into a single branck of $7 \times 7$ convolution Ding et al. (2021). Furthermore, Hu et al. proposed online convolutional re-parameterization (OREPA) to leverage a linear scaling at each branch to diversify the optimization directions, instead of applying non-linear normalization after convolution layer Hu et al. (2022). Also, stack of small kernels are leveraged to generate similar receptive field of view as LKs with better training and inference efficiency. The effectiveness of leveraging small kernels stack and multiple parallel branches design will be further investigated as another directions of our future work.

