# OpenReview forum: "3D UX-Net: A Large Kernel Volumetric ConvNet Modernizing Hierarchical Transformer for Medical Image Segmentation"
_ICLR.cc/2023/Conference — ICLR 2023 poster_

### Official Review · Reviewer_8E6F · 2022-10-23

**Confidence:** 5
**Correctness:** 3
**Technical Novelty And Significance:** 3
**Empirical Novelty And Significance:** 4
**Recommendation:** 8

**Clarity, Quality, Novelty And Reproducibility:**

As mentioned above the paper is clear, of high quality and reproducible. The novelty is high and comes from a mixture of empirical and methodological findings. The impact and the interest of the community in performant alternatives to both nnUNet and SwinUNETR is arguably high (looking at the large number of medical segmentation challenges e.g. at MICCAI).

**Strength And Weaknesses:**

**Strengths**
+ the paper is well written, easy to follow and addresses a relevant task in 3D/medical image analysis.
+ the experimental section contains numerous direct comparisons with different versions of Swin/Vision-Transformers and demonstrates both faster convergence and higher accuracy of the proposed 3D UX-Net
+ the authors also evaluate pre-train/fine-tune scenarios and reach SOTA performance on AMOS22
+ within the method a new module: "depthwise convolutional scaling (DCS)" is introduced that performs slightly better than MLPs

the work gives (yet another) reason to consider claimed advantages of transformers with much care, since e.g. the increased receptive field of using 7x7x7 convolutions together with reasonable practical design choices alone can outperform those in a simpler Conv-only model

**Weaknesses**
- while closely following the narrative of ConvNeXt (which I liked) the authors perform there ablation study in the reverse direction: ie. in Tab 3 they start from the proposed model and **remove** individual parts, while ConvNeXt starts from a plain ResNet and **adds** the transformer-inspired changes. This is unfortunate, because for 3D segmentation there is indeed one prominent framework the nnUNet that uses a plain Conv-only model and has to date won nearly every medical segmentation challenge (incl. AMOS22). This is also evident from Tab. 1 where the nnUNet outperforms every single transformer approach and is only beaten by the proposed method (in light of these facts I would also strongly recommend to rephrase/remove the statement "To our best knowledge, this is the first 3D ConvNet architecture that competes favorably with transformers SOTA in volumetric segmentation tasks.)  A somewhat more elaborate discussion on the differences of the nnUNet (which e.g. also doesn't use BatchNorm) would be of important value. Importantly: the kernel-size ablation should in addition be extended by 3x3x3 and 5x5x5!
- I am also missing a more detailed discussion about runtimes for training and inference, since the raw FLOPs numbers are not always very favourable for UX-Net and depth-separable convolutions are additionally prone to lower efficiency (when comparing similar FLOP numbers). On the other hand the nnUNet is known to be fairly resource intensive for training, so any method that reduces this burden would be of great impact!
As a minor comment (since this was published after the ICLR submission deadline): one could include the nnUNet results for AMOS22 from https://arxiv.org/pdf/2208.10791.pdf in the final version

**Summary Of The Paper:**

The paper explores the necessity of incorporating transformer elements (self-attention, swin module) into 3D segmentation networks. It demonstrates SOTA performance on three challenge datasets (FLARE, FeTA, AMOS) with comparable model size and computational FLOPS. It is heavily inspired Liu's "A ConvNet for the 2020s" CVPR paper, but nonetheless makes a meaningful contribution for practical 3D applications that could be further enhanced by a slightly better ablation study.

**Summary Of The Review:**

A recommend acceptance, subject to moderate improvements in terms of the comprehensiveness of the ablation study. Both the practical impact and the methodological insights (revisiting larger kernels and depth-separable convolutions in contrast to transformer modules) are relevant.

---

> ### Author Response · Authors · 2022-11-17
> **Response to Reviewer 8E6F**
>
> Thank you reviewer to have such constructive comments towards our work, here is our thoughts to further clarify and modified the details in our revised manuscripts:
>
> > while closely following the narrative of ConvNeXt (which I liked) the authors perform there ablation study in the reverse direction: ie. in Tab 3 they start from the proposed model and remove individual parts, while ConvNeXt starts from a plain ResNet and adds the transformer-inspired changes.
>
> - Thank you for the valuable comments! We agree with the reviewer to a small extent that we start from the swin transformer blocks in SwinUNETR and add the transformer-inspired changes using CNN modules to replace both self-attention and MLP modules. The ablation studies in Tab. 3 have provided step-by-step guidance on how we adapt our innovative thoughts using CNN modules, starting from different kernel sizes to the effectiveness of MLP.
>
> > This is unfortunate, because for 3D segmentation there is indeed one prominent framework the nnUNet that uses a plain Conv-only model and has to date won nearly every medical segmentation challenge (incl. AMOS22). This is also evident from Tab. 1 where the nnUNet outperforms every single transformer approach and is only beaten by the proposed method (in light of these facts I would also strongly recommend to rephrase/remove the statement "To our best knowledge, this is the first 3D ConvNet architecture that competes favorably with transformers SOTA in volumetric segmentation tasks.) A somewhat more elaborate discussion on the differences of the nnUNet (which e.g. also doesn't use BatchNorm) would be of important value. Importantly: the kernel-size ablation should in addition be extended by 3x3x3 and 5x5x5!
>
> - For the CNN-based SOTA nn-UNet, we agreed that the nn-UNet uses a plain Conv-only model to demonstrates its effectiveness in different medical segmentation challenges. We additionally discuss the performance comparison and the network architecture of nnUNet in Appendix A.3. We believe that such performance improvement is mainly contributed to **their proposed self-configured training strategies and the ensembling postprocessing, instead of the network itself (plain 3D U-Net only)**. To further verify our network performance, we substitute the 3D U-Net architecture in nn-UNet with our proposed network and demonstrate another significant improvement from 0.934 to **0.944** (FLARE 2021) and 0.874 to **0.881** (FeTA 2021) with p<0.01 respectively. We have also additionally added the ablation studies of 3x3x3 and 5x5x5 kernels in the latest version of our manuscripts.
>
> - We have further modified the statement “To our best knowledge, this is the first 3D ConvNet architecture that competes favorably with transformers SOTA in volumetric segmentation tasks.” as follows:
> > To our best knowledge, this is the first block design of leveraging 3D depthwise convolutions to competes favorably with transformer SOTA in volumetric segmentation tasks
>
> ---
>
> > I am also missing a more detailed discussion about runtimes for training and inference, since the raw FLOPs numbers are not always very favourable for UX-Net and depth-separable convolutions are additionally prone to lower efficiency (when comparing similar FLOP numbers). On the other hand the nnUNet is known to be fairly resource intensive for training, so any method that reduces this burden would be of great impact!
> - Thank you for the comments on runtimes for training and inference. We further add a discussion section in Appendix A.4 and discuss about the FLOPs in 3D UX-Net. To reduce the burden of low efficiency, we found that withdrawing the bottleneck layer and increasing the depth of stages (e.g., 8 blocks) further reduced the FLOPs from 639.4G to **536.3G**, while keeping current performances. Another possible solution is to adapt reparameterization with consecutive small kernel convolution layers (e.g., 3 3x3x3 convolution = 1 7x7x7 convolution receptive field) [1,2,3,4], which demonstrates to reduce FLOPs for both training and inference in 2D large kernel CNNs and will be our future focus for 3D tasks.
>
> > As a minor comment (since this was published after the ICLR submission deadline): one could include the nnUNet results for AMOS22 from https://arxiv.org/pdf/2208.10791.pdf in the final version
> - For the nnUNet performance in AMOS 2022, we have further integrated it into the latest version of manuscript. Following our finetuning scenario, our proposed network 3D UX-Net still outperforms nn-UNet with 2.51% of mean organ Dice (nnUNet: 0.878, 3D UX-Net: 0.900). Here, we only train the nn-UNet for 250 epochs (40000 steps ~ 250 epochs) instead of 1000 epochs as default setting to ensure the fair comparison for the network convergences.

---

> > ### Author Response · Authors · 2022-11-17
> > **References in the Responses**
> >
> > References:
> >
> > 1. Ding, Xiaohan, et al. "Scaling up your kernels to 31x31: Revisiting large kernel design in cnns." Proceedings of the IEEE/CVF Conference on Computer Vision and Pattern Recognition. 2022.
> > 2. Ding, Xiaohan, et al. "Re-parameterizing Your Optimizers rather than Architectures." arXiv preprint arXiv:2205.15242 (2022).
> > 3. Ding, Xiaohan, et al. "Repvgg: Making vgg-style convnets great again." Proceedings of the IEEE/CVF Conference on Computer Vision and Pattern Recognition. 2021.
> > 4. Hu, Mu, et al. "Online Convolutional Re-parameterization." Proceedings of the IEEE/CVF Conference on Computer Vision and Pattern Recognition. 2022.

---

### Official Review · Reviewer_ERdC · 2022-10-24

**Confidence:** 3
**Correctness:** 4
**Technical Novelty And Significance:** 2
**Empirical Novelty And Significance:** 3
**Recommendation:** 8

**Clarity, Quality, Novelty And Reproducibility:**

The study is well communicated with a high standard of clarity and reproducibility.
The novelty of the paper may be questioned as many design choices for the proposed model have been examined separately or in part.

**Strength And Weaknesses:**

Strength:
The manuscript is well written and constructed, the figures and tables are coherent.
The provided context in the literature is adequate.
The experimental validation is well applied and provides evidence to support the authors conclusions.

Weaknesses:
While the authors provide an extensive literature review to provide the right context to their research, it would be beneficial if they could explicitly outline in paragraph 2.2 the differences of the proposed approach to the previously published models. Particularly, relating to the main inspiration of this paper, e.g. Liu et al. 2022, in the context of novelty should be better clarified.


**Summary Of The Paper:**

The authors present an extension to convolutional networks (ConvNets) tailored to simulate the behavior of hierarchical transformers, aiming to provide an alternative to visional transformers (ViT) with a significantly reduced number of trainable parameters. The architectural variations in the proposed model focused on volumetric depth-wise convolutions with large kernel size for feature extraction that enables larger global receptive field. They then provide experimental evidence using 3 datasets that demonstrated an overall improvement in performance of 3D semantic segmentation in medical imaging.

**Summary Of The Review:**

In the current stage of scientific research in medical image segmentations, after the immersion of Visual transformers, this study provides a relevant point of view, aiming to simulate a convolutional neural network alternative with comparable performance and simplified architecture. The study is well applied and communicated in this manuscript and would be a good contribution to the field and ICLR.

---

> ### Author Response · Authors · 2022-11-18
> **Response to Reviewer ERdC**
>
> Thank you for the constructive comments towards our manuscripts. We further integrated your comments and revised our manuscripts as follows:
>
> > While the authors provide an extensive literature review to provide the right context to their research, it would be beneficial if they could explicitly outline in paragraph 2.2 the differences of the proposed approach to the previously published models. Particularly, relating to the main inspiration of this paper, e.g. Liu et al. 2022, in the context of novelty should be better clarified.
>
> - We further refine the paragraph 2.2 to specify the differences between our proposed approach and the previous works as follows:
> > Apart from transformer-based framework, previous works began to revisit the concept of depthwise convolution and adapt its characteristics for robust segmentation. It has been proved to be a powerful variation of standard convolution that helps reduce the number of parameters and transfer learning. Zunair et al. introduced depthwise convolution to sharpen the features prior to fuse the decoded features in a UNet-like architecture. 3D U$^2$-Net leveraged depthwise convolutions as domain adaptors to extract domain-specific features in each channel. **Both studies demonstrate the feasibility of using depthwise convolution in enhancing volumetric tasks. However, only a small kernel size is used to perform depthwise convolution.** Several prior works have investigated the effectiveness of large kernel (LK) convolution in medical image segmentation. For instance, Huo et al. leveraged LK (7x7) convolutional layers as the skip connections to address the anatomical variations for splenomegaly spleen segmentation; Li et al. proposed to adapt LK and dilated depthwise convolutions in decoder for volumetric segmentation. **However, significant increase of FLOPs is demonstrated with LKs and dramatically reduces both training and inference efficiency.** To enhance the model efficiency with LKs, Liu et al. proposed ConvNeXt as a 2D generic backbone that simulate ViTs advantages with LK depthwise convolution for downstream tasks with natural image, while ConvUNeXt is proposed to extend for medical image segmentation and compared only with CNN-based networks (e.g. ResUNet, UNet++). **However, limited studies have been proposed to efficiently leverage depthwise convolution with LKs in a volumetric setting and compete favorably with volumetric transformer approaches. With the large receptive field brought by LK depthwise convolution, we hypothesize that LK depthwise convolution can potentially emulate transformers' behavior and efficiently benefit for volumetric segmentation.**
>
> We further highlight and bold specific sentences to further specify the novelty and motivations of our work in the latest version of manuscript.

---

> ### Author Response · Authors · 2022-12-07
> **Follow up with the response!**
>
> Dear Reviewer ERdC,
>
> As the deadline of the discussion period is coming along, we want to make sure that our response have addressed all your concerns, and we will appreciate very much if you could give us any feedback. In particular, we have refined the literature review, specifically for paragraph 2.2, to further outline our motivation and novelty of our work. We further adapt our network with nn-UNet training strategies, which further boost the segmentation performance from 0.934 to 0.944 mean Dice (In Appendix A.3). We hope that our responses will resolve your main concern and also provided more insights about our paper in your review. We are also more than happy to answer your further questions. Thank you very much for your time and efforts!

---

### Official Review · Reviewer_Ud7h · 2022-10-26

**Confidence:** 4
**Correctness:** 2
**Technical Novelty And Significance:** 2
**Empirical Novelty And Significance:** 1
**Recommendation:** 3

**Clarity, Quality, Novelty And Reproducibility:**

The paper is clearly written, however as mentioned above is not sufficiently well motivated. The reading of the results is problematic based on single runs and minor improvements. This could have implications on reproducibility.


**Strength And Weaknesses:**

**Strengths:**

* The authors propose a volumetric segmentation method, that is evaluated on multiple relevant datasets. The choice of different architectural components in this work are explained clearly.

* Ablation studies are reported and model hyperparameters are reported clearly.

* Experiments on multiple datasets and several relevant baselines are conducted.

**Weaknesses:**

* **Motivation for the model**: This work aims to reintroduce CNNs to match the performance of Vision transformers (ViTs) for volumetric medical image segmentation. This is strange to me, as the chronology of how these models were developed is somehow warped. This work claims to revisit CNNs for volumetric segmentation, and improve them so that they can emulate ViTs -- which were already mimicking CNNs. This logic of arguments is circular to me. I found it very hard to subscribe to the primary motivation of this work. Further, and more fundamentally, the claim that ViTs have surpassed CNNs in volumetric segmentation is highly exaggerated (first sentence in abstract). Going by even the results reported in this work this claim does not hold up.

* **Experimental evaluation**: Results in Table 1 which have the main results are reported in single repeats. For the scale of performance improvements that is claimed, this could be just random, as the difference in Dice accuracy is very small (first or second decimal in most cases). Reading them as improvements based on single model runs is highly problematic. These small differences could simply be overcome with different initialisations. Also, the number of parameters between the current model and a CNN baseline like nn-Unet is double. The claims of superior performance under these conditions is not convincing.

* **Additional insights**: Unfortunately, there are no new insights gained from this work. How does the use of the depth-wise convolution with large kernels influence segmentation across different tasks. What is the reason for these improvements? What type of errors that the other ViT or CNN models make are alleviated with this new architecture? If ViTs were capable of modelling longe range interactions and CNNs introduced image-specific inductive biases, how does the new model bridge this gap? How can this be evaluated?

**Summary Of The Paper:**

Volumetric segmentation of medical images is still an open problem. This work proposes a new U-net type segmentation network that uses large receptive field depth-wise convolution operations to emulate vision transformers (ViTs). Additionally, patch-wise feature propagation and 1x1 convolutions are introduced as means of reducing the compute compared to ViTs. Experiments on multiple volumetric datasets and relevant CNN and ViT baseline models are reported with small to no improvements.

**Summary Of The Review:**

This work proposes a volumetric segmentation method with a motivation of bridging the performance gap between ViTs and CNNs, by adding depth-wise convolutions with large kernels and couple of other tricks. The premise of the work that ViTs have surpassed CNNs, and CNNs need to emulate ViTs is flawed. Experimental evaluation shows small or no improvements, even with orders of magnitude more compute/parameters compared to other baseline methods.

---

> ### Author Response · Authors · 2022-11-18
> **Response to Reviewer Ud7h (1/2)**
>
> Thank you reviewer to consider as clearly written and provide the detailed comments towards our work, here is our thoughts to further clarify both motivations and insights with additional experiments as follows:
>
> **Motivation for the model**
> > This work aims to reintroduce CNNs to match the performance of Vision transformers (ViTs) for volumetric medical image segmentation. This is strange to me, as the chronology of how these models were developed is somehow warped. This work claims to revisit CNNs for volumetric segmentation, and improve them so that they can emulate ViTs -- which were already mimicking CNNs. This logic of arguments is circular to me. I found it very hard to subscribe to the primary motivation of this work.
> -  We apologize for the confusion and hope to explain our motivation and logic more clearly in the rebuttal response. For recent 3D ViTs (e.g. UNETR [7], nnFormer [8]) in medical domain, it is challenging to adapt 3D ViT models for high-resolution dense prediction tasks due to the high complexity of computing global self-attention with respect to the input size. The hierarchical transformers (e.g., SwinUNETR [1,2]) tackle such limitations and achieved the superior performance compared with CNN benchmarks in volumetric segmentation. **Such performance gain is largely owing to the larger field of view (FOV) from 3D shift window multi-head self-attention (MSA), which is computationally unscalable to achieve via traditional 3D volumetric CNN architectures.** The recent advance in depthwise convolution-based large convolutional kernel design (e.g., Liu et al [3]) provides a computationally scalable mechanism for larger FOV in 2D CNNs. Inspired by the large FOV in 3D ViTs and scalable large convolutional kernel design, this study revisits the 3D volumetric CNN design to investigate the feasibility of **(1) achieving the SOTA performance via a pure CNN architecture**, **(2) yielding much less network complexity compared with 3D ViTs**, **(3) providing a new direction of designing 3D CNN networks on volumetric data**. Briefly, we hypothesis that introducing the large kernel convolution to volumetric CNN can largely mimic the advantages of window-based MSA and shifted-window MSA in 3D ViTs (e.g., SwinUNETR [2,3]).
>
> > “Further, and more fundamentally, the claim that ViTs have surpassed CNNs in volumetric segmentation is highly exaggerated (first sentence in abstract). Going by even the results reported in this work this claim does not hold up.”
> - First, we agree with the reviewer that such an expression (e.g., the first sentence in the abstract) is too exaggerated. We have modified the first sentence as follows:
> > The recent 3D ViTs (e.g., SwinUNETR) achieve state-of-the-art performances on several 3D volumetric data benchmarks, including 3D medical image segmentation.
>
> - **From the recent studies**, the ViTs outperform CNN with different training scenarios (e.g., large-scale pretraining and finetuning) [4,5,6]. UNETR [7], nnFormer [8] and SwinUNETR [2,3] claimed that their performance outperforms nnUNet in their scenarios.
> - **From Table 1 in this paper**, the performance of nn-UNet is mainly contributed by **the self-configuring training strategies and predictions ensembling postprocessing, instead of its network structure (plain 3D U-Net, first row in Table 1)**. To benchmark different datasets, our goal is to evaluate the ability of network architecture only. The transformer-based approaches, especially SwinUNETR, outperform most of the plain network architectures (e.g., 3D U-Net, SegResNet) and hierarchical CNN-based architectures (e.g., RAP-Net).
>
> **Experimental Evaluation**
> > Results in Table 1 which have the main results are reported in single repeats. For the scale of performance improvements that is claimed, this could be just random, as the difference in Dice accuracy is very small (first or second decimal in most cases). Reading them as improvements based on single model runs is highly problematic. These small differences could simply be overcome with different initializations.
> - For the difference in Dice accuracy, we want to clarify that all results are computed as an average across rigor nested **5-fold cross-validations** in training FeTA 2021 and FLARE 2021 datasets Moreover, such differences are statistically significant with Wilcoxon signed-rank test (p<0.01). Furthermore, **from Table 2**, the finetuning performance of 3D UX-Net outperforms SwinUNETR for **2.27 % of mean organ Dice (0.880 to 0.900) across 15 organs**. Our proposed 3D CNN achieves a better performance compared with SOTA 3D ViTs, **with significantly less network parameters**.

---

> > ### Author Response · Authors · 2022-11-18
> > **Response to Reviewer Ud7h (2/2)**
> >
> > Continue with Experimental Evaluation:
> > > Also, the number of parameters between the current model and a CNN baseline like nn-Unet is double. The claims of superior performance under these conditions is not convincing.
> > - For the claim about the CNN-baseline nn-UNet, we found that the structure of our proposed network 3D UX-Net can be further optimized to reduce the model parameters by eliminating the bottleneck layer and increasing the depth of 3D UX-Net blocks (e.g., 8 blocks) in stage 3, while demonstrating similar performance with the current proposed structure. Such network architecture further reduces both model parameters (from 53.0M to **32.1M**, nn-UNet: 31.2M) and FLOPs (from 639.4G to **536.8G**, nn-UNet: 743.3M). However, such findings are already beyond the scope of this work, and we further put a discussion section in **Appendix section A.4**. In the quantitative performance aspect, as the network architecture in nn-UNet is **a plain 3D U-Net (Table 1, first row)**, we further substitute the network with 3D UX-Net and demonstrate consistent performances of mean organ Dice from 0.934 to **0.944** (FLARE 2021) and from 0.874 to **0.881** (FeTA 2021) with **p<0.01** respectively.
> >
> > **Additional Insights**
> > > “How does the use of the depth-wise convolution with large kernels influence segmentation across different tasks?”
> > -  Adapting large kernel depthwise convolutions provide large receptive fields to extract features and enhance the correspondence within regions, which is similar to the self-attention mechanism in ViTs. We evaluate the depthwise convolution effectiveness on **multi-modality imaging (CT, MRI) for multi-organ segmentation**. **From table 2 & 3, we quantitatively found that the improvement of segmentation performance depends on different kernel sizes, which proportional to the volumetric field of view in different datasets (FeTA 2021: small brain infant scans, FLARE 2021 / AMOS 2022: full body / abdominal scan)**. For more fine-grain theoretical explanations towards such improvement, we agree that it is vitally important to consider as the context of explainable AI, while it is out of scope from our paper’s motivation.
> >
> > > “What type of errors that the other ViT or CNN models make are alleviated with this new architecture?”
> > - Thank you for raising an important point. **As shown in Figure 4**, we found that ViTs yield inaccurate segmentation towards the neighboring regions at a high qualitative level, for both training from scratch and finetuning scenarios. The errors are appeared to be on preserving the fine-grained boundary information of organ of interests. Our proposed architecture further demonstrates its robustness and stability in segmenting fine-grain details across multi-modality imaging.
> >
> > > “If ViTs were capable of modelling long range interactions and CNNs introduced image-specific inductive biases, how does the new model bridge this gap? How can this be evaluated?”
> > - We agree with the reviewer that the evaluation of theoretical models is exceptionally important. In this paper, we have presented an empirical approach characterizing the validity of simulating transformer behaviors using CNN modules for volumetric segmentation, thus further emphasize that deeper theoretical considerations are necessary. This work shows that there is empirical evidence of a knowledge gap for spanning deeper considerations in the context of explainable AI, which is crucially important.
> >
> > References:
> > 1. Hatamizadeh, Ali, et al. "Swin unetr: Swin transformers for semantic segmentation of brain tumors in mri images." International MICCAI Brainlesion Workshop. Springer, Cham, 2022.
> > 2. Tang, Yucheng, et al. "Self-supervised pre-training of swin transformers for 3d medical image analysis." Proceedings of the IEEE/CVF Conference on Computer Vision and Pattern Recognition. 2022.
> > 3. Liu, Zhuang, et al. "A convnet for the 2020s." Proceedings of the IEEE/CVF Conference on Computer Vision and Pattern Recognition. 2022
> > 4. Liu, Zhaoshan, and Lei Shen. "Medical image analysis based on transformer: A Review." arXiv preprint arXiv:2208.06643 (2022).
> > 5. Shamshad, Fahad, et al. "Transformers in medical imaging: A survey." arXiv preprint arXiv:2201.09873 (2022).
> > 6. Li, Jun, et al. "Transforming medical imaging with Transformers? A comparative review of key properties, current progresses, and future perspectives." arXiv preprint arXiv:2206.01136 (2022).
> > 7. Hatamizadeh, Ali, et al. "Unetr: Transformers for 3d medical image segmentation." Proceedings of the IEEE/CVF Winter Conference on Applications of Computer Vision. 2022.
> > 8. Zhou, Hong-Yu, et al. "nnformer: Interleaved transformer for volumetric segmentation." arXiv preprint arXiv:2109.03201 (2021).
> > 9. Isensee, Fabian, et al. "nnU-Net: a self-configuring method for deep learning-based biomedical image segmentation." Nature methods 18.2 (2021): 203-211.

---

> > > ### Comment · Reviewer_Ud7h · 2022-12-12
> > > **Response to author responses**
> > >
> > > I thank the authors for their clarifications. Specifically, for pointing out that the results reported were based on five fold cross validation. I find it strange the authors do not report these cross-validated measures with standard deviations.
> > >
> > > With regards to the other two concerns, I appreciate the new content in Appendix A.4; I actually think this is the more interesting result from this work. As this is primarily an empirical work, showing improvements in compute/parameters could have strengthened the claims in the work.
> > >
> > > However, as the key contribution is the large kernel depth-wise convolution, the narrative in this work might not be sufficiently interesting to the volumetric/medical imaging community. Also, the role of how this model is bridging ViTs and CNNs is still unclear. So, I choose to keep my score.

---

> > > > ### Author Response · Authors · 2022-12-12
> > > > **Thank you for the feedbacks. Response to your comments**
> > > >
> > > > We are glad that our further clarifications and the new added contents have strengthened the claims in our work, especially for the two concerns related to the motivations and the additional insights with more interesting result by adapting 3D UX-Net with nn-UNet training strategies in the revised version. We further provide response to your further comments:
> > > >
> > > > > Specifically, for pointing out that the results reported were based on five fold cross validation. I find it strange the authors do not report these cross-validated measures with standard deviations.
> > > >
> > > > For 5-fold cross validations, the testing split is divided into 5 non-overlapping combinations of samples to minimize the overfitting of our model. Standard deviations cannot be computed across folds (as we are not using the same testing samples for all 5-folds training) and we agree that we can show the standard deviations across all 5-folds testing subjects. However, our revised version has further added the latest performance of adapting our network in nn-UNet training strategies and have demonstrated large improvements in multi-organ segmentation with multi-modality images datasets (FeTA 2021: **0.870 (nn-UNet) to 0.881 (3D UX-Net)**; FLARE 2021: **0.929 (SwinUNETR) to 0.944 (3D UX-Net)**, **p<0.01, Wilcoxon signed-rank test**).
> > > >
> > > > > However, as the key contribution is the large kernel depth-wise convolution, the narrative in this work might not be sufficiently interesting to the volumetric/medical imaging community. Also, the role of how this model is bridging ViTs and CNNs is still unclear.
> > > >
> > > > For further clarification, our key contribution is not the large kernel depth-wise convolution, as the large kernel depth-wise convolution have been widely used in natural image domain and demonstrates to learn features with large receptive field in an efficient manner. Once again, our key contributions consists of three folds: **(1) achieving the SOTA performance via a pure ConvNet architecture**, **(2) yielding much less network complexity compared with 3D ViTs**, and **(3) providing a new direction to simulating hierarchical transformer behavior with new 3D ConvNet block design on volumetric high resolution tasks**. Our work is to provide another explorative thought to extract features with large receptive field efficiently and propose a new scaling technique in volumetric setting for segmentation task, instead of bridging the gap between ViTs and CNNs. **We hypothesis that certain characteristics of the intrinsic structure of 3D transformer can be simulated by pure 3D CNN modules**.
> > > >
> > > > Previous works have been demonstrated to use large kernel convolution in decoder block and compare 2D large kernel depthwise convolution with CNN network baseline only (e.g., Attention U-Net, ResUNet, UNet++) [1,2]. To our best knowledge, **we propose the first 3D block design with large kernel depth-wise convolution in encoder network to simulate transformer behavior and compete favorably with current transformer SOTA approaches for volumetric segmentation**.
> > > >
> > > > We hope that our responses will resolve your further concerns and are also more than happy to answer your further questions. If our response resolves your concerns, we kindly ask you to re-consider raising the rating of our work. Thank you very much for your time and efforts!
> > > >
> > > >
> > > > Reference:
> > > > 1. Li, Hao, et al. "Large-Kernel Attention for 3D Medical Image Segmentation." arXiv preprint arXiv:2207.11225 (2022).
> > > > 2. Han, Zhimeng, Muwei Jian, and Gai-Ge Wang. "ConvUNeXt: An efficient convolution neural network for medical image segmentation." Knowledge-Based Systems 253 (2022): 109512.

---

> ### Author Response · Authors · 2022-12-02
> **Your feedback is important to us**
>
> Dear Reviewer Ud7h,
>
> This is a friendly reminder that we have submitted our response to your review comments and uploaded a revision of the paper two weeks ago, and we will appreciate very much if you could give us any feedback. In particular, our response provides more detailed clarifications in the motivation and additional insights of our work. We further add ablation studies using our network with nn-UNet training strategies, which further boost the segmentation performance from 0.934 to 0.944 mean Dice. We hope that our responses will resolve your main concern and also provided answers to other questions you asked in your review. If our response resolves your concerns, we kindly ask you to consider raising the rating of our work. We are also more than happy to answer your further questions. Thank you very much for your time and efforts!

---

> ### Author Response · Authors · 2022-12-07
> **Follow up to the responses!**
>
> Dear Reviewer Ud7h,
>
> As the discussion period will be ended within a week, your feedback is vitally important towards our work and we want to ensure that our response have addressed all your concerns. Our work demonstrates an empirical approach characterizing the validity of simulating transformer behaviors using CNN modules for volumetric segmentation. Also, we further added clarifications in both motivation and additional insights of our work, and ablation studies with nn-UNet training strategies to further demonstrate our network ability. Meanwhile, we completely agree to your comments that deeper theoretical considerations are necessary. If our response resolves your concerns, we kindly ask you to consider raising the rating of our work. If you have any further questions, we will definitely reply it as soon as possible.

---

### Author Response · Authors · 2022-11-18
**General Response**

We thank all reviewers for their constructive comments and suggestions regarding our work. We appreciate all reviewers to consider our work to be well written. We further appreciate the enthusiasm from reviewer ERdC and reviewer 8E6F: "well communicated with a high standard of clarity and reproducibility" and "provides a relevant point of view, aiming to simulate a convolutional neural network alternative with comparable performance and simplified architecture" (reviewer ERdC). Furthermore, " The novelty is high and comes from a mixture of empirical and methodological findings" (reviewer 8E6F).

After carefully reading comments and suggestions from all reviewers, we have made the following new contributions and changes:
- We refined the introduction section to further clarify the motivation and insights of our work (reviewer Ud7h).
- We refined the related work section for depthwise convolution-based segmentation to outline the main inspiration and clarify our novelty (reviewer ERdC).
- We ran additional experiments to evaluate our proposed network in nn-UNet architecture and further demonstrate an additional mean organ Dice improvement with significance (p<0.01) using our network design (Appendix A.3). We further added a discussion section to discuss about the architectural difference between the plain 3D U-Net in nn-UNet and our proposed network (reviewer Ud7h and reviewer 8E6F). We also included the nn-UNet performance comparison for AMOS 2022 dataset in finetuning scenario. Our proposed network demonstrates consistent improvement compared to all state-of-the-art CNN-based and Transformer-based approaches in all scenarios (reviewer 8E6F).
- We ran additional ablation studies with different kernel sizes (3x3x3 and 5x5x5) and further investigated the network structure to reduce the FLOPs. An additional discussion section is added to further elaborate the findings from ablation studies and the possible directions (e.g., re-parameterization) to reduce the burden of low efficiency with large kernels (reviewer 8E6F).

All changes are available in the revised manuscript with highlighted in yellow.

We want to thank all the reviewers again and the area chair for their valuable time and hope that our comments can address all your concerns. We are happy and available to have further discussions throughout the rebuttal period.

---

### Decision · Program_Chairs · 2023-01-20

**Decision:**

Accept: poster

**Justification For Why Not Higher Score:**

While two out of three reviewers strongly recommend acceptance, one reviewer still questions the significance of the work to the volumetric biomedical image analysis community.

**Justification For Why Not Lower Score:**

There is good agreement that the method is novel and leads to tangible benefits in terms of accuracy, computational complexity, and parameter count.

**Metareview: Summary, Strengths And Weaknesses:**

Inspired by the performance of vision transformers, the authors of this paper re-design convnet architecture for volumetric image analysis leading to improvements in accuracy, parameter count, and computational complexity over SOTA.

**Note From Pc:**

if the above contains the word "oral" or "spotlight" please see: "oral" presentation means -> notable-top-5% and "spotlight" means -> notable-top-25%. As stated in our emails, we are disassociating presentation type from AC recommendations